# Maximising the Use of Scarce qPCR Master Mixes

**DOI:** 10.3390/ijms23158486

**Published:** 2022-07-30

**Authors:** Stephen Bustin, Claire Bustin, Sara Kirvell, Tania Nolan, Reinhold Mueller, Gregory Shipley

**Affiliations:** 1Medical Technology Research Centre, Anglia Ruskin University, Chelmsford CM1 1SQ, UK; sara.kirvell1@aru.ac.uk (S.K.); tanianolan@btinternet.com (T.N.); 2Royal Wolverhampton NHS Trust, Wolverhampton WV10 0QP, UK; c.bustin1@nhs.net; 3RM Consulting, San Diego, CA 92037, USA; reinholdmueller7@gmail.com; 4Shipley Consulting, Vancouver, WA 98682, USA; gshipley14@me.com

**Keywords:** PCR, qPCR, sensitivity, PCR efficiency, molecular diagnostics

## Abstract

The COVID-19 pandemic resulted in a universal, immediate, and vast demand for comprehensive molecular diagnostic testing, especially real-time quantitative (qPCR)-based methods. This rapidly triggered a global shortage of testing capacity, equipment, and reagents. Even today, supply times for chemicals from date of order to delivery are often much longer than pre-pandemic. Furthermore, many companies have ratcheted up the price for minimum volumes of reaction master mixes essential for qPCR assays, causing additional problems for academic laboratories often operating on a shoestring. We have validated two strategies that stretch reagent supplies and, whilst particularly applicable in case of scarcity, can readily be incorporated into standard qPCR protocols, with appropriate validation. The first strategy demonstrates equivalent performance of a selection of “past expiry date” and newly purchased master mixes. This approach is valid for both standard and fast qPCR protocols. The second validates the use of these master mixes at less than 1x final concentration without loss of qPCR efficiency or sensitivity.

## 1. Introduction

The polymerase chain reaction (PCR) is the most ubiquitous molecular technology in use today. There are tens of thousands of laboratories carrying out research applications that rely on PCR, most commonly in a real-time (qPCR) [1] or digital (dPCR) [2] format. PCR-based testing has also long been the mainstay of accurate, fast and high throughput diagnostic testing for infectious diseases [3]. Hence, there are numerous companies manufacturing and supplying individual components or complete kits required to carry out PCR assays and diagnostic tests. Nevertheless, one of the unpredicted corollaries of the current COVID-19 pandemic has been the apparent scarcity of reagents as well as the infrastructure needed to carry out those tests [4,5]. This has resulted in a variety of strategies aimed at increasing the testing capacity for SARS-CoV-2, including drawing on assistance from research laboratories [6], developing alternative sampling protocols [7], combining reagents sets [8], reducing primer and probe concentration [9] and advocating faster cycling times [10]. However, apart from shortening denaturation and polymerisation times, which can be readily implemented, all other modifications require some degree of change and revision of protocols for pathogen type-specific testing. In this communication we propose two universal and unanticipated adaptations that are useful not just for diagnostic testing but can be implemented for any qPCR-based experiment. Our proof of principle experiments demonstrate that it is possible to use qPCR master mixes long past their expiry dates, even with PCR amplicons as long as 437 bp. We also show many master mixes can be used at much less than the conventional 1x final concentration. 

## 2. Results

### 2.1. Past-Expiry Date Master Mixes

The benchmark PCR efficiencies of the four assays were established using an “unexpired” IDT master mix and were around 100% (Figure 1C). The three “past-expiry date” ABI probe master mixes MM-1, MM-2 and MM-3 were compared to one newly purchased, MM-4, using a standard PCR protocol. This consisted of a 2-min activation step, followed by 40 cycles of denaturation (5 s at 95 °C) and polymerisation (20 s at 60 °C). PCR amplicons amplified by assays A and D were detected by LNA hydrolysis probes, those amplified by assays B and C by DNA probes. Amplification plots and resulting Cqs recorded were similar for the four assays (Figure 2A–D), with the shortest PCR amplicon (assay D) recording the lowest Cq. The comparability of master mix was also shown by the average Cq values and low standard deviations across the four master mixes at 25.64 ± 0.3, 26.22 ± 0.29, 25.35 ± 0.26 and 24.48 ± 0.35 for assays A, B, C and D, respectively. ∆Cq values of each of the “past expiry date” master mixes relative to that of MM-4 supported the observation that there was no discernible difference in the performance of the four master mixes (Figure 2E). There also was no marked difference between the LNA (assays A and D) and DNA (assays B and C) probes.

The experiment was repeated with a fresh preparation of primers and probes. This time the annealing/polymerisation time was reduced to 10 s. Amplification plots and Cqs were similar to those shown in Figure 2 (Appendix A). Assay D again recorded the lowest Cq values; all ∆Cq values relative to MM-4 were well within experimental noise levels, confirming that all master mixes amplified the four assays with approximately equal efficiency (Appendix A). 

Next, sufficient premixes were prepared to carry out a series of experiments using progressively reduced denaturation and polymerisation times, starting at 5 s denaturation/10 s polymerisation and ending at 1 s denaturation/1 s polymerisation. A comparison of the ∆Cq values recorded by each of the “past expiry date” master mixes relative to the Cq values obtained with the fresh master mix again emphasised the comparability of each one’s performance (Appendix A). Even the longest assay A, at 437 bp, was efficiently amplified under the shortest amplification conditions (Appendix A). These results were not confined to amplification and analysis on one instrument only. When the four assays were amplified with MM-1 to MM-4 and detected on three different qPCR instruments, comparable results were achieved (Appendix A). 

Similar amplification results were obtained with all four master mixes when target DNA concentrations were increased 50-fold (Appendix A). A repeat amplification and analysis with just MM-2 on a different qPCR instrument gave the same result (Appendix A). Noticeably though, the amplification plots of assays A, B and C are flatter than those of assay D, indicating less efficient amplification during the later stages of the PCR reaction.

The performances of the expired master mix labelled “Advanced Fast Master mix”, MM-1 and that of the current version with the same designation, MM-4, were further analysed by comparing PCR efficiencies for the longest assay, A. First, a reference standard curve was generated with Bioline SensiFast master mix, and recorded an efficiency of 91.4% (Figure 3A), similar to the one obtained with the IDT master mix (Figure 1C). MM-4 master mix was similarly efficient at 94.9% and recorded slightly lower Cqs (Figure 3B). The efficiency of MM-1 was assessed twice, with comparable results (Figure 3C,D). 

Limits of detection between the two master mixes were compared by amplifying ten replicates of the highest dilution standard with MM-1 and MM-4 on the PCRMax Eco instrument. A 1-min activation was followed by 40 cycles of a 5 s denaturation step at 95 °C and a ten second polymerisation step at 60 °C. MM-1 amplified 3/10 replicates, whereas MM-4 amplified 4/10. Consequently, both master mixes were equally sensitive at their limits of detection, which for this assay was between 1 and 10 copies. 

Next, a “past expiry date” probe master mix, from a different supplier, BioRad’s iQ Supermix (MM-6), was used to amplify the four PCR amplicons on the BioRad CFX using a medium fast 1-min activation, 5 s denaturation/15 s polymerisation protocol. The run recorded equivalent Cq values with assays C and D, slightly higher Cqs with assay B and barely amplified the longest PCR amplicon A (Appendix A). Increasing the polymerisation time to 60 s improved both the shape of the amplification plots and resulted in a lower Cq for assay A (Appendix A). The experiments were repeated on the PCRMax Eco. Increasing the polymerisation times from 15 s (Appendix A) to 30 s (Appendix A) and 60 s (Appendix A) incrementally improved the amplification results. However, on this instrument, even at the 60 s polymerisation time, assay A was not amplified efficiently. 

The four assays were also amplified with the ABI SYBR Green mix (MM-5) using a 1-min activation, 1 s 95 °C/20 s 60 °C protocol. The longest assay, A, amplified the least well, and the shortest assay, D, the best (Figure 4A). Assays B and C recorded similar Cqs. Melt curve analysis showed single melt curves for assays C and D. Assay A recorded two peaks, whereas assay B had a leading shoulder (Figure 4B). Additionally, noticeable were the low ∆Rn values for assay A.

The poorer melt curve patterns of the two longest assays A and B are probably associated with a the less optimal, short amplification protocol used in this experiment, as it was ameliorated by using longer polymerisation times (data not shown).

Given these results, the experiment was repeated with probes spiked into the reactions. Amplification patterns were comparable to those achieved with SYBR Green, except that Cq as well as ∆Rn values were higher for all four assays (Figure 4C). When the experiment was repeated with the polymerisation time extended to 1 min, all four assays recorded similar results (Figure 4D). 

### 2.2. Master Mix Dilutions

Assay D was amplified with twelve master mixes using a conventional PCR protocol (3-min activation. Followed by 40× [5 s at 95 °C/60 s at 60 °C]. Master mixes were used at a standard 1×, as well as reduced 0.8, 0.7 and 0.5× concentrations. Surprisingly, amplification plots were similar (Appendix A) and the master mixes recorded similar Cq values regardless of master mix concentration (Figure 5A–C). Consequently, the experiment was repeated with 1× and 0.4× concentration final master mix reactions. This time the ∆Rn values were lower with all master mixes, but the Cq values remained surprisingly resilient (Figure 5D and Appendix A). It was only at 0.3× final master mix concentration that these master mixes stopped amplifying efficiently (data not shown). 

Five master mixes (Promega, IDT, PCRBio, ABI (MM-1) and BioLine) were chosen for further analysis. First, all four assays were amplified with the Promega master mix using a 5 s denaturation/15 s polymerisation protocol on the BioRad CFX. The ∆Cq values recorded by the four assays at the 0.5× and 0.4× master mix concentration indicated that the master mix performed comparably well at all three concentrations (Figure 6A), and well within experimental noise range. Results were similar when the experiment was repeated with a new batch of oligonucleotide primers and probes (Figure 6B). 

The amplification efficiency of the Promega master mix with assay D was assessed by running two standard curves, one at 0.5×, the other at 0.4× concentration of master mix using the same 5 s denaturation/15 s polymerisation protocol. Amplification efficiencies at both concentrations were approximately equivalent at 93.1% (Figure 6C) and 89.2% (Figure 6D) for amplification carried out at 0.5× and 0.4× final concentrations, respectively. Limits of detection were determined for longest assay A by amplifying twelve replicates of one of the dilution standards used earlier for the PCR efficiency determinations. The runs were carried out on the PCRMax Eco using the 5 s denaturation/15 s polymerisation protocol. At 1× final concentration, 10/12 replicates recorded Cq values, as did 9/12 and 10/12 for the 0.5× and 0.4× final concentrations, respectively (Figure 6E). These results indicated that reducing the concentration of the master mix did not affect the sensitivity of the assays. 

Finally, amplifications of assay A were carried out using MM-1 and a range of MM-1 aliquots that had been subjected to five, ten, fifteen and twenty cycles of thawing and freezing. The slopes of the amplification plots differed slightly between the stock and frozen/thawed aliquots and final fluorescence levels were lower for the frozen/thawed master mixes. (Figure 7A). However, Cq values for the five samples were similar at 21.33 ± 0.14, 22.00 ± 0.08, 21.83 ± 0.14, 21.92 ± 0.19 and 21.93 ± 0.19, respectively. Whilst the control master mix recorded the lowest Cq vales, the differences in Cq (∆Cq) were not statistically significant (Wilcoxon test, two-tailed *p* = 0.0625), being between 0.3 and 0.5 higher for the frozen/thawed samples (Figure 7B). When the experiment was repeated with 0.5× final concentrations of each aliquot, results were similar, with endpoint fluorescene levels again lower (Figure 7C). Cq values were also comparable (21.14 ± 0.15, 21.58 ± 0.05, 21.36 ± 0.08, 21.59 ± 0.11 and 21.31 ± 0.06) as were ∆Cq values (Figure 7D), with differences between Cq values again not significant. 

## 3. Discussion

One of the apparent barriers to early universal testing at the start of the current COVID-19 pandemic was a localised lack of supplies that hampered diagnostic laboratory operations and the inability to increase testing capacity [4,5]. The exact nature of these shortages remains unclear but, anecdotally and from our own experience, we have encountered delays to the ordering of PCR reagents and plastic ware: hence the need for both diagnostic and research laboratories to stretch stocks. Two obvious ways of doing this is to use reagents past their expiry dates or use them at less than the standard 1× concentration. Despite the hiatus during the pandemic, such considerations are generally less of an issue for testing laboratories with high reagent turnover. However, they can be an issue for research laboratories, especially smaller ones carrying out less frequent PCR analyses. Uncertainty brought on by uncertain supply and stability of reagents past their expiry dates prompts the question of whether old reagents should be discarded “to be on the safe side”, and so waste precious resources and funds or risk using them, fingers crossed. Such decisions are unlikely to be mentioned in the Materials section of any subsequent publication. Manufacturers and suppliers warn against this practice, but this is unsurprising as it is in their interest to sell as much product as possible for as high a price as possible. Whilst there’s a lot of advice available online, there is very little peer-reviewed information available that would help clarify this issue. One report demonstrates that diagnostic antibodies have a workable half-life in excess of 10 years [11] and that repeated cycles of freezing and thawing of plasma do not affect a range of immunological assays [12]. Given reagents in a PCR master mix must be sufficiently robust to withstand multiple cycles of extreme heating and cooling, it seems reasonable to assume that such master mixes are at least as tough as antibodies or the proteins contained in plasma. There is no information at all about whether it is possible to use less than the standard 1× *g* concentration of master mix. 

The results shown in Figure 2 and Appendix A provide clear evidence that it is patently safe to use master mixes that are considerably past their expiry date. PCR amplicons ranging in length from 437 bp to 153 bp were amplified equally by an unexpired master mix compared to three that were years past their expiry dates. We deliberately chose to stress the reaction by amplifying a range of long PCR amplicons that might be expected to be amplified poorly with expired reagents. Optimally, qPCR assays are much shorter than any of these. Two important issues to note are (i) the same results were observed when conventional, slow PCR cycling times were used or when they were reduced to medium fast (10 s polymerisation, Appendix A) and maximum speed (1 s polymerisation, Appendix A) on a conventional qPCR instrument and (ii) results were reproducible across three different types of qPCR instruments (Appendix A). These were (i) a conventional instrument with a reduced mass sample block heated and cooled by Peltiers, (ii) an instrument built around a thermal block filled with circulating conductive fluid and (iii) a rotary-based instrument that achieves heating and cooling by magnetic induction. Samples containing high concentrations of the four assay targets were amplified equally well by all master mixes (Appendix A). Amplification efficiencies and limits of detection were equivalent, providing further proof of the suitability of these past expiry date reagents for both diagnostic and research use (Figure 3).

BioRad’s iQ Supermix, which expired in 2016, did not perform as well under the medium fast (15 s polymerisation) PCR cycling conditions. However, increasing the polymerisation time to 60 s improved both the shape of the amplification plots and reduced the Cq values recorded by assay A (Appendix A). The inference that the BioRad *Taq* polymerase and/or buffer did not support fast PCR cycling conditions was confirmed when the experiments were repeated on the PCRMax, which has much faster ramp rates (Appendix A). With a 15 s polymerisation time, only the shortest assay D performed well. At 30 s, there was only a small improvement in Cq values recorded by assay D, whereas assay C recorded much lower Cqs. Assay B did not amplify efficiently until the polymerisation time was increased to 60 s. Even with the longer time assay A still not amplified efficiently. These results suggest that the poorer performance of this master mix is not a consequence of its age. Instead, these performance issues may be due to the characteristics of the *Taq* polymerase and/or buffer components. These include the length of time needed to activate the enzyme, its processivity or its speed. It does, however, emphasise the importance of testing and validating the performance of different enzymes and reagents carefully prior to use. 

The SYBR Green master mix (MM-5) performed well with assays B, C and D using a 20 s polymerisation step protocol, but assay A performed less well (Figure 4A). Amplification with assays C and D resulted in single melt curves, whereas assay B showed a leading shoulder and assay A had two peaks (Figure 4B). As we do not know how the Taq polymerase or buffers differ between the probe-based and SYBR Green assays, we are unable to explain the difference in performance. Spiking in a probe and repeating the experiment, gave approximately the same result as above, with assay A performing a little less well (Figure 4C). Extending the polymerisation time to 60 s restored the performance of all four assays detected by probes (Figure 4D). One thing that was notable was the low ∆Rn values recorded with SYBR Green, which could be a possible indicator of problems with past expiry date SYBR Green master mix. 

We also tested the concentration of master mix required to generate reliable, sensitive results that are equivalent to the ones achieved with standard 1x concentration reagents. Again, this is an unanticipated issue that has never been discussed in the peer-reviewed literature. An evaluation of a number of master mixes demonstrated that all could be used for real-time detection at 0.8× (Figure 5A) and several could be used at 0.5× (Figure 5C) or even 0.4× (Figure 6D) final concentration. Results were reproducible, and the use of reduced master mix concentration as low as 0.4× was perfectly feasible even with “past expiry” master mixes, such as the one from BioRad or MM-1. PCR efficiencies at 0.5× and 0.4× final concentration were comparable to the 1× concentration, as were the limits of detection (Figure 6). This is likely due to the fact that the Cq value characteristic of real-time detection is determined during the early exponential phase of the PCR reaction where all master mix components, even when diluted, remain abundant. However, with some master mixes there was a marked difference between real-time and end-point detection of PCR amplicons (Appendix A), extending to all master mixes at 0.4× final concentration (Appendix A). This is likely to be due to limiting enzyme or dNTP concentrations, exacerbated by the dilution, and resulting in fewer PCR amplicons being synthesised during the late stages of the PCR reaction. It is also possible that, additionally, the variation in endpoint values with diluted reagents is compounded by the accumulation of end products, inhibitors and inactivated polymerases. This association between reduced final master mix concentration and lower final fluorescence value indicates that the use of diluted master mixes for endpoint assays, for example those used for genotyping, is not advisable. 

Finally, we investigated whether the number of freeze thaw cycles rather than time past expiration might be an important factor. We chose MM-1, as it had been kept frozen since being aliquoted once in 2014 and subjected fresh aliquots to a range on freeze/thaw cycles. The results indicate that, at least for this master mix, freeze/thawing is not a major factor in performance reduction (Figure 7). Whilst the slopes of the original stock and the frozen/thawed aliquots differed slightly, Cqs were comparable. Intriguingly, whilst the amplification patterns were similar for the reactions run at 1× and 0.5× final concentrations, Cqs for all aliquots were lower at the latter concentration. Nevertheless, we would suggest that users aliquot their master mixes and do not subject them to too many cyles of freeze/thawing.

In summary, with proper validation and appropriate protocols and instruments, master mixes years past their expiry dates can generate qPCR results that are as specific, sensitive and reliable as those generated by newly purchased ones. Whether this extends to the use of other PCR-based methods such as dPCR remains to be determined and independently validated. Consequently, using reagents past their expiry date or at a lower than recommended concentration is a useful recommendation especially in a research context where resources can be scarce and the research budget tight. It may be less attractive for diagnostic laboratories, except in exceptional circumstances such as the current COVID-19 pandemic. Clearly, it is always important to validate both master mix and assay as well as report the modification as part of any publication of the data. An additional corollary of our findings is that they are another indicator of how our understanding and use of PCR is very much underexplored and that many of our assumptions and recommendations [13] would benefit from an update and revision.

## 4. Materials and Methods

### 4.1. Reagents and qPCR Instruments

The details of the master mixes used for the “past expiry date” experiments are shown in Figure 8. All had been stored at −20 °C since arrival in the laboratory, the first one since 2013 (MM-5), the most recent one since March 2022 (MM-4). MM-1 (2014) and MM-2 (2016) had been thawed and refrozen once, whereas MM-3 (2014), MM-5 (2013) and MM-6 (2015) had been thawed and refrozen numerous times.

Details of additional master mixes and the three qPCR instruments used are listed in Figure 1A. 

Assays were designed to target the genomic DNA (gDNA) of the fungal pathogen *Candida auris*. The gDNA sequence specifying the internal transcribed spacer 1, 5.8S ribosomal RNA gene, internal transcribed spacer 2 and large subunit ribosomal RNA gene of *C. auris* (OK001825.1) was downloaded to the Beacon Designer qPCR assay design software package (Premier Biosoft, San Francisco, CA, US). Primers were designed with manual adjustments aimed at obtaining PCR amplicon sizes ranging from very long to moderately long by qPCR standards (Figure 1B). Two forward and three reverse primers were chosen to amplify four sets of PCR amplicons, which could be detected using two probes, one being DNA only, the other a DNA/LNA probe. PCR amplicons varied in length from 437 bp (assay A) to 153 bp (assay D) (Figure 1C). 

The specificity of primers, probes and amplicons was analysed in silico using Primer-BLAST (https://www.ncbi.nlm.nih.gov/tools/primer-blast/, accessed on 4 April 2022) and BLAST (https://blast.ncbi.nlm.nih.gov/Blast.cgi, accessed on 4 April 2022). Upon receipt, all DNA oligonucleotides were resuspended in sterile RNase-free water at 100 µM and stored in aliquots at −20 °C. *C. auris* gDNA was a gift to the laboratory from Prof C. Lass-Flörl, Medizinische Universität Innsbruck, Austria. 

### 4.2. Protocols

#### 4.2.1. PCR Efficiencies of Assays A to D

The benchmark PCR efficiencies of assays A to D were determined using IDT’s PrimeTime qPCR master mix. Standard curves for the four assays were prepared by carrying out serial 10-fold dilutions of previously amplified PCR amplicons in water. 2× pre-mixes of *C. auris* gDNA, primers, probes and water were prepared for each of the four assays and added to equal volumes of the 2× IDT master mix. Primers were used at 500 nM, probes at 200 nM final concentration. Reagents were mixed by pipetting, briefly centrifuged, and placed on ice. 5 µL reaction volumes were amplified on a PCRMax Eco qPCR instrument using a protocol involving a 1-min activation step at 95 °C, followed by 40 cycles of 5 s at 95 °C and 20 s at 60 °C. Quantification cycle (Cq) values were determined using default threshold and baseline settings.

#### 4.2.2. Analysis of “Past Expiry Date” Probe-Based Master Mixes

The amplification and detection characteristics of MM-1 to MM-4 and MM-6 master mixes were analysed by preparing 2× pre-mixes of *C. auris* gDNA, primers, probes and water for each of the four assays and adding the premixes to equal volumes of each of the 2× master mixes. Primers were used at 500 nM, probes at 200 nM final concentration. Reagents were mixed by pipetting, briefly centrifuged, and placed on ice. PCR reactions were carried out in 5 µL volumes on BioRad CFX Connect and PCRMax Eco qPCR instruments or 10 µL volumes on a BMS Mic qPCR instrument. Apart from the first experiment, which was carried out using a conventional PCR protocol (2 min activation, 40 cycles of 5 s at 95 °C and 20 s at 60 °C), the PCR reactions were run under faster conditions, generally using 1 s cycling times. Details are listed in the text below as well as the legends of the appropriate figures. Cq values were determined using threshold and baseline settings suggested by the instrument software. 

The PCR efficiency of assay A with a “past expiry date” master mix was assessed using MM-1. Reactions were carried out on the same run as control reactions using two “in date” master mixes: ABI’s current master mix (MM-4) and an additional “in date” comparator master mix (Meridian Bioscience London UK, SensiFast BIO-86005). 5 µL reaction volumes were amplified on a PCRMax Eco qPCR instrument using a protocol involving a 1-min activation step at 95 °C, followed by 40 cycles of 5 s at 95 °C and 10 s at 60 °C. Denaturation and polymerisation times were kept longer than equivalent ones on the BioRad CFX because of the faster ramp rates of the PCRMax Eco resulted in less time spent between temperatures. Cq values were determined using default threshold and baseline settings.

#### 4.2.3. Analysis of “Past Expiry Date” SYBR Green Master Mix

2x pre-mixes of *C. auris* gDNA, primers and water were prepared for each of the four assays and added to equal volumes of 2× MM-5 SYBR master mix. Primers were used at a final concentration of 300 nM or at 500 nM if probes (200 nM final concentration) were spiked into the master mix. Reagents were mixed by pipetting, briefly centrifuged, and placed on ice. PCR reactions were carried out using 5 µL volumes on a PCRMax Eco qPCR instrument. Apart from a conventional PCR protocol (2 min activation, 40 cycles of 5 s at 95 °C and 20? cycles at 60 °C), PCR reactions were also run under a variety of conditions, details of which are listed in the text below and legends of the appropriate figures. Cq values were determined using default threshold and baseline settings.

#### 4.2.4. Validation of the Use of Diluted Master Mixes

Premixes of *C. auris* gDNA, primers and water were prepared for each of the dilutions. These were added to appropriate amounts of 2× master mixes to generate final master mix concentrations of 1x and below. Primers were used at 500 nM, probes at 200 nM final concentration. Reagents were mixed by pipetting, briefly centrifuged and placed on ice. PCR reactions were carried out using 5 µL volumes on BioRad CFX Connect or PCRMax Eco qPCR instruments. PCR protocols varied and are described in the text below and in the figure legends. Cq values were determined using default threshold and baseline settings.

Dilution curves for assay A were obtained by preparing two premixes of Promega master mix, one at 0.5×, the other at 0.4× final concentration, plus primers (500 nM), probes (200 nM) and water. The premixes were dispensed into two series of microfuge tubes, to which serial 10-fold dilutions of the PCR amplicon were added. 5 µL reaction volumes were amplified on a BioRad CFX qPCR instrument using a protocol involving a 3-min activation step at 95 °C, followed by 40 cycles of 5 s at 95 °C and 15 s at 60 °C. Cq values were determined using default threshold and baseline settings.

#### 4.2.5. Effects of Thawing and Freezing

The impact of repeated thawing/freezing was investigated using MM-1, purchased as a 50 mL stock in 2014. MM-1 had been thawed once to aliquot some of the contents and the remainder was kept frozen ever since. Aliquots were deliberately thawed and refrozen five, ten, fifteen and twenty times over an eight-week period. The original MM-1 stock and the frozen and thawed aliquots were used to amplify assay A on the BioRad CFX. Primers were used at 500 nM, probes at 200 nM final concentration and reactions were carried out in 5 µL volumes on a BioRad CFX instrument. A fast protocol was used that comprised a 1-min activation step followed by 40 cycles of 1 s at 95 °C and 1second at 60 °C. The different master mix samples were used at 1× *g* as well as at 0.5× *g* final concentration.

#### 4.2.6. Data Analysis

All data were initially analysed using the software supplied with each instrument, then imported and further analysed in Microsoft Excel for Mac v.16.61 and PRISM for Mac v.9.3.1.

## Figures and Tables

**Figure 1 ijms-23-08486-f001:**
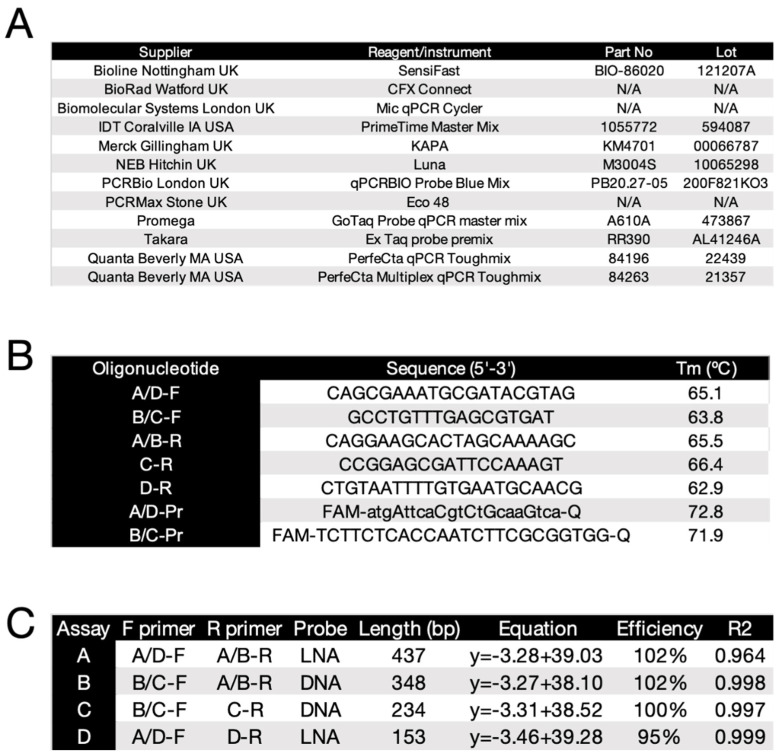
Reagents, instruments, oligonucleotides and assays for dilution experiments. (**A**) Suppliers and materials/instruments used in this report; (**B**) Oligonucleotide sequences. The capital letters in oligonucleotide A/D-Pr designate LNA bases; (**C**) Details of the four assays. PCR efficiencies were established using IDT’s PrimeTime master mix.

**Figure 2 ijms-23-08486-f002:**
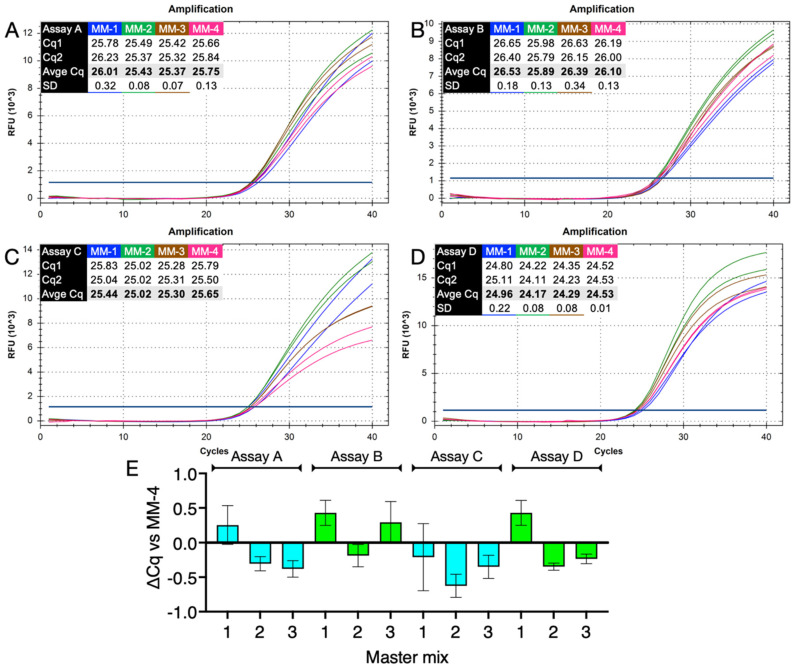
Amplification by MM-1 to MM-4 using standard PCR conditions (2 min activation at 95 °C, followed by 40 cycles of 5 s at 95 °C denaturation and 20 s at 60 °C annealing/polymerisation steps). MM-1 (blue), MM-2 (green), MM-3 (brown), MM-4 (pink). Assays A and D were detected by an LNA probe, assays B and C by a DNA probe. (**A**). Amplification plots and Cq values assay A (437 bp). (**B**). Amplification plots and Cq values assay B (348 bp). (**C**). Amplification plots and Cq values assay C (234 bp). (**D**). Amplification plots and Cq values assay D (153 bp). (**E**). ∆Cq values (±SD) for past expiry date MM-1 to 3 relative to the current MM-4.

**Figure 3 ijms-23-08486-f003:**
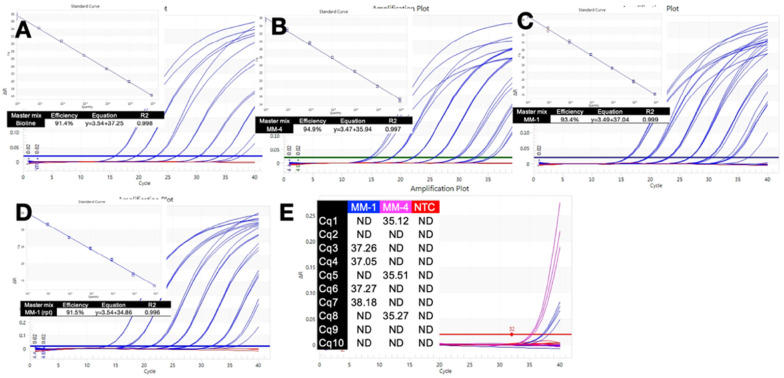
**Comparison of PCR efficiencies.** (**A**). Amplification plots, standard curve and assay parameters recorded with Bioline master mix. Red plots indicate the NTCs. (**B**). Amplification plots, standard curve and assay parameters recorded with MM-4. Red plots indicate the NTCs. (**C**). Amplification plots, standard curve and assay parameters recorded with MM-1 master mix. Red plots indicate the NTCs. (**D**). Amplification plots, standard curve and assay parameters recorded for the repeat reaction of the standard curve experiment with MM-1. Red plots indicate the NTCs. (**E**). Limits of detection for MM-1 (blue) and MM-4 (pink) using the highest dilution standard.

**Figure 4 ijms-23-08486-f004:**
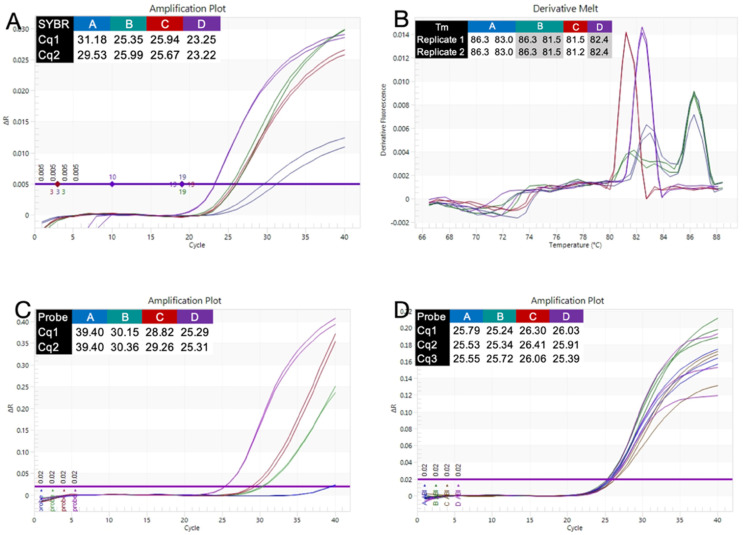
Amplification of assays A (blue), B (green), C (brown) and D (purple) with MM-5. (**A**). Cq values and amplification plots recorded with MM-5 master mix with a 20 s polymerisation step. (**B**). Tms and melt curves for each of the four assays. Assay A has two distinct peaks and assay B has a leading shoulder. (**C**). Cq values and amplification plots following the addition of hydrolysis probes with a 20 s polymerisation step. (**D**). Cq values and amplification plots following the addition of hydrolysis probes with a 60 s polymerisation step.

**Figure 5 ijms-23-08486-f005:**
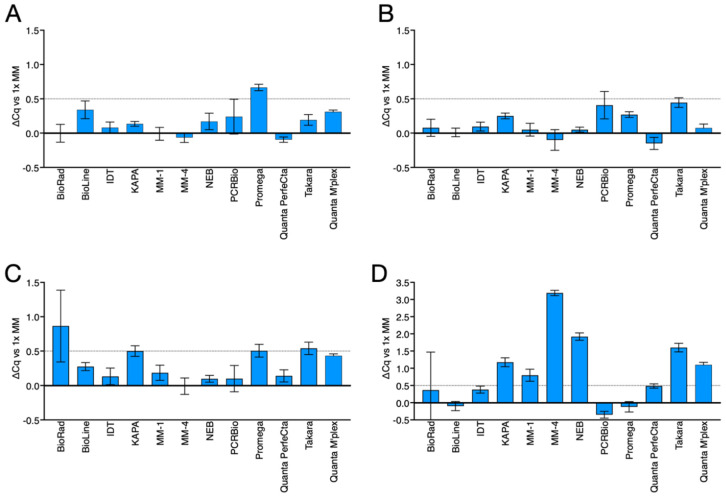
Comparison of amplifications carried out at reduced concentration of master mix. (**A**). ∆Cq values for the twelve master mixes at 0.8× concentration. (**B**). ∆Cq values for the twelve master mixes at 0.7× concentration. (**C**). ∆Cq values for the twelve master mixes at 0.5× concentration. (**D**). ∆Cq values for the twelve master mixes at 0.4× concentration. The dotted line indicates a ∆Cq value of 0.5 relative to 1× concentration of master mix.

**Figure 6 ijms-23-08486-f006:**
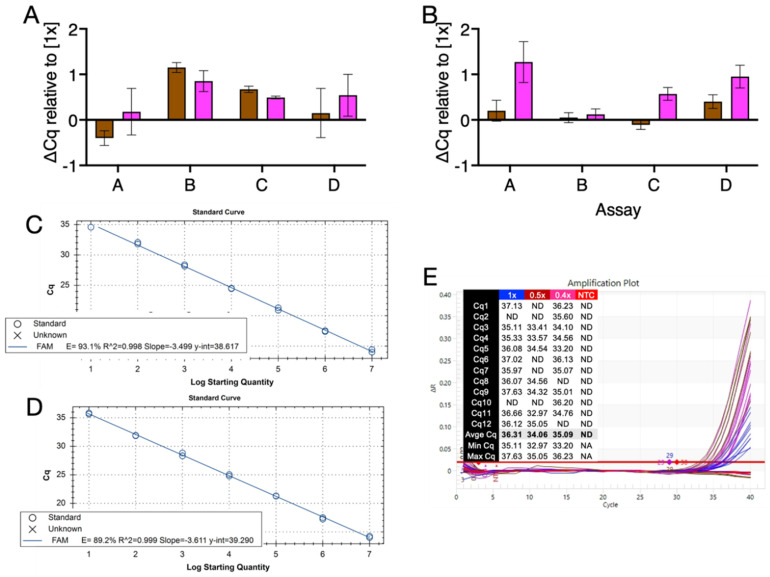
Assessment of Promega master mix at various final concentrations. (**A**). ∆Cq values of 0.5× (brown) and 0.4× (pink) compared with 1× final concentration recorded with the four assays. (**B**). Repeat reaction with freshly prepared primers and probes. ∆Cq values of 0.5× (brown) and 0.4× (pink) compared with 1× concentration recorded with the four assays (**C**). Standard curve and PCR efficiency of assay D amplified by 0.5× concentration master mix. (**D**). Standard curve and PCR efficiency of assay D amplified by 0.4× concentration master mix. (**E**). Limits of detection for assay A at 1× (blue), 0.5 (brown) or 0.4× (pink) master mix concentration.

**Figure 7 ijms-23-08486-f007:**
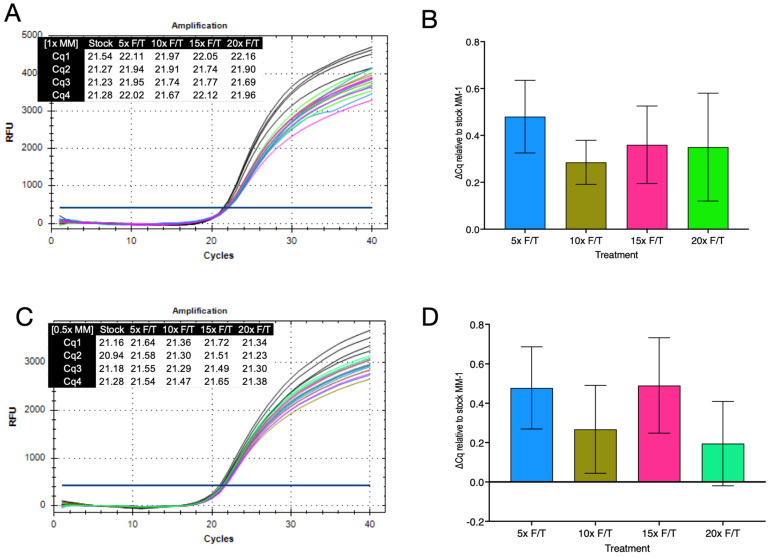
Analysis of the effect of repeated cycles of freezing and thawing (F/T) on MM-1. Original MM-1 stock (black), aliquots frozen and thawed 5 times (blue), 10 times (olive), 15 times (pink) and 20 times (green). (**A**). Amplification plots and Cq values for assay A amplified with 1× final concentrations of master mix. (**B**). ∆Cq values relative to stock MM-1 of the various MM-1 aliquots amplified at 1× final concentration. (**C**). Amplification plots and Cq values for assay A amplified with 0.5× final concentrations of master mix. (**D**). ∆Cq values relative to stock MM-1 of the various MM-1 aliquots amplified at 0.5× final concentration.

**Figure 8 ijms-23-08486-f008:**
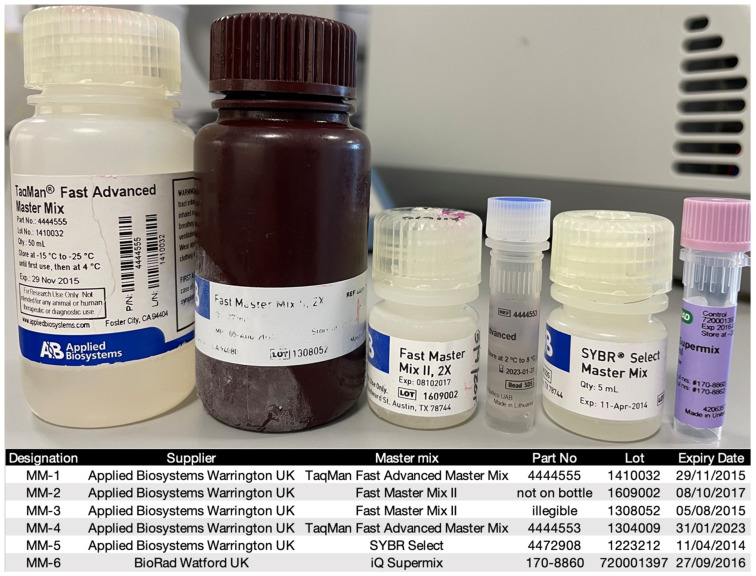
The six master mixes tested for past expiry date performance, with expiration dates.

## Data Availability

All data supporting the results reported here are included with Figures.

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
