# Peer review of "Maximising the Use of Scarce qPCR Master Mixes"

_ijms, 2022, doi:10.3390/ijms23158486_

Round 1

Reviewer 1 Report

The manuscript "Maximising the use of scarce PCR master mixes" evaluates the use of a variety of commercial qPCR kits at concentrations below, and times exceeding the expiry dates, of those recommended by the manufacturers. This work is a direct response to the difficulties experienced by many researchers as a result of "supply-chain" shortages experienced during the COVI-19 pandemic. The authors conduct a series of well-reasoned, empirical tests, and find compelling evidence that such commercial kits can provide reliable results: when extended to testing twice as many samples (than the number recommended), and/or when used several years after their advertised expiry date.

Obviously, it is not in the interest of the manufacturers to conduct or publish such work (because they want to maximize their sales), and so this kind of independent study is extremely valuable. The manuscript was clear, concise, and a pleasure to read. I think it will prove to be a highly-cited publication.

I have little to offer in the way of criticism other than to highlight a few spelling corrections -

Page 5, line 138, "times" rather than "timeds"?

Page 6, line 191, "was" rather than "as"?

Page 8, line 261, "prior" rather than "propr"?

One final comment - out of interest, was their any particular reason why you used LNA probes in two of your assays and DNA probes in the other two? I understand that LNA probes are supposedly more specific (because they typically work at higher temperatures), but was that a consideration in your experimental design?

Reviewer 2 Report

I would like to thank for the opportunity to review this paper.

This paper focused on the usage of PCR master mixes and tested the activities of qPCR master mixes long past their expiry dates, and low concentration of master mixes, claiming that using master mixes which have expired or at a lower than recommended concentration is a useful strategy when needed. Another indicating from this paper is that freeze/thawing may not influence the the performance of the master mixes tested in this paper.

However, many issues need to be further addressed.

1. Background: The authors should introduce more about the PCR, such as reverse transcription-PCR, digital PCR,  and real-time PCR (qPCR). What are the difference between them, and why this paper focused on qPCR. Depend on the purpose, the requirements for specific, sensitive, and reliable results by PCR are different. However, for research or diagnostic purposes, PCR need to be more specific and sensitive, which may not fit for the usage of expired or a low concentration master mixes.

2. The authors tested some PCR master mixes in this paper, which can not cover all PCR master mixes in the market. Even the same product with different lot numbers may differ from each other. So the data provided here may not be sufficient for support the conclusion that expired or a lower than recommended concentration master mixes are suitable for research usage.

3. Cq value is used for the evaluation of the quality of the PCR master mixes, however, as the authors mentioned in line 282-283, it only reflect the early exponential phase of the PCR reaction. Such as in Figure 1C, the Cq of MM-4 is similar to the others while the curves show more differences. Hence, Cq may not be a good parameter for the quality of PCR master mixes.

4. The expired master mixes listed in Figure 7 are well maintained in -20 degree fridge. However, whether those reagents stored in 4 degree (reagents in use) also remain high quality for PCR reaction?

5. For the dilution testing of master mixes, as shown in Figure S6 and S7, Cq values can not evaluate the activities of master mixes and the curves show different performance according to the dilution folds. Besides, different reagents behave significant differently indicating the results here may not be applied to all PCR master mixes.

Minor errors:

1. Figure 7, MM-1 expiry date is 29 Nov, 2015.

2. Line 261, "propr to use", is it "prior to use"?

Round 2

Reviewer 2 Report

The authors addressed some of my concerns, but still some concerns remain to be addressed.

1. The authors should also make it clear in the title and abstract that this paper focuses only on qPCR instead of PCR  and discusses how qPCR is used to determine reliably whether there is measurable pathogen in a sample or not using expired or lower concentration qPCR master mixes. Although I see the authors made some changes.

2. The background should be improved to introduce more about the qPCR: its development, applications, advantages and disadvantages; and illustrate the importance and  significance of this research. As known, all of the reagents are not designed to be used after expiry date or at a lower concentration.

3. Both MM-5 and MM-6 are expired which need fresh reagents as controls.

4. In figure 6A and 6B, why aliquots frozen and thawed 10 times (orange) shows better quality?

5. the color labeling should be improved in some of the figures: Figure 6A and 6C, Figure S3.
